# Effect of Cr on the Mineral Structure and Composition of Cement Clinker and Its Solidification Behavior

**DOI:** 10.3390/ma13071529

**Published:** 2020-03-26

**Authors:** Haihong Fan, Mengqi Lv, Xiaosha Wang, Jianmin Xiao, Xiaofan Mi, Luwei Jia

**Affiliations:** College of Materials Science and Engineering, Xi’an University of Architecture and Technology, Xi’an 710055, China; lvmeng_qi@163.com (M.L.); xiaosha0309@126.com (X.W.); xiaomin0622@126.com (J.X.); mxf0912@126.com (X.M.); jlw0925@126.com (L.J.)

**Keywords:** clinker, heavy metal, solidification, ^29^Si MAS NMR, Cr

## Abstract

In order to reveal the solidification behavior of Cr in the cement clinker mineral phase, ^29^Si magic-angle spinning nuclear magnetic resonance, X-ray diffraction, and scanning electron microscopy with energy-dispersive X-ray spectroscopy techniques were used to analyze the morphology and composition of the cement clinker mineral phase doped with Cr. The results showed that the addition of Cr did not change the chemical environment of ^29^Si in the clinker mineral phase, and it was still an isolated silicon–oxygen tetrahedron. Cr affected the orientation of the silicon–oxygen tetrahedron and the coordination number of calcium, leading to the formation of defects in the crystal structure of the clinker mineral phase, by replacing Ca^2+^ into the mineral phase lattice to form a new mineral phase Ca_3_Cr_2_(SiO_4_)_3_. Cr acted as a stabilizer for the formation of β-C_2_S in the clinker calcination. As the amount of Cr increased, the relative content of C_3_S decreased and the relative content of C_2_S increased. Further, Cr easily dissolved in C_2_S, while it was not found in C_3_S. This study is conducive to further research on the mechanism of heavy metal solidification in cement clinker. Furthermore, it is important to evaluate the environmental risk of heavy metals in the process of sludge disposal through cement kiln and promote the utilization of sludge resources and the sustainable development of the cement industry.

## 1. Introduction

As of the end of June 2018, the amount of municipal and industrial sludge in China reached 90 million tons, with a utilization rate of only 30%, and random landfills have seriously polluted the environment. The main reason for restricting its recycling is that it contains heavy metals. The refractory and enrichment of heavy metals hinders the sludge resource utilization. At present, the coprocessing technology of cement kilns is recognized as one of the safest and most effective sludge disposal methods in the world. By utilizing its advantages such as high temperature in the kiln, long residence time of materials, and stable incineration status, dangerous substances such as heavy metal ions in the sludge can be solidified in the mineral phase of cement clinker [1,2,3]. Heavy metal ions will form an enriched phase due to the crystal structure defect of the cement clinker mineral phase itself, and will also be present in the empty interstitial spaces of the mineral phase [4,5,6,7,8]. Therefore, the research on the influence of heavy metal ions on the calcination degree of clinker and the mineral phase structure of clinker has gained attention.

Chromium is one of the heavy metal elements often present in sludge. The influence of chromium on clinker quality must be considered in the safe and environmentally friendly production of chromium-containing sludge in the coprocessing of cement kilns. Scientists all around the world have carried out significant research on this. Katyal [9] found that when the content of Cr_2_O_3_ is less than 2%, T-type C_3_S is formed. When the Cr_2_O_3_ content increased to 4–5%, the C_3_S crystal form changes to the M1 form, leading to the growth of a new mineral phase CaCrO_4_. However, according to Enculescu [10], when the content of Cr_2_O_3_ is 0.5%–2.0%, the crystal form of C_3_S changes to the M1 type. At the same time, when the content of Cr_2_O_3_ is 4%, a new phase, CaCrO_4_, is also formed. Xi and Kolovo [11,12,13] found that Cr does not exist alone in the form of compounds during the clinker calcination. It exists mainly in the form of Cr^3+^ and Cr^5+^ which replace Si^4+^ in silicate minerals. Barros [14] showed that the incorporation of Cr in the clinker can inhibit the formation of C_2_S, while Fierens [15] believed that doping Cr can prevent the formation of C_3_S in the clinker. Stephan [16,17] pointed out that when Cr is added at a low concentration (0.02%–0.5%), it has no effect on the formation of C_3_S. Only at a high content (greater than 1.0%), Cr will decompose C_3_S to generate more C_2_S and CaO. Thus, previous studies on the crystal structure change, generation quality of clinker mineral phases doped with Cr, and role of Cr in the reduction of the stability of C_3_S or C_2_S are not consistent [18].

In this study, chromium nitrate nonahydrate was used for doping cement clinker. Techniques such as ^29^Si magic-angle spinning nuclear magnetic resonance (MAS NMR), X-ray diffraction (XRD), and scanning electron microscopy with energy-dispersive X-ray spectroscopy (SEM-EDS) were used to analyze the cement clinker doped with Cr from the point of view of mineral phase structure, content, and mineral phase formation and morphology. The solidification behavior of Cr in the mineral phase of cement clinker and the degree of its influence on the formation of the mineral phase were investigated. The aim is to provide a new theoretical basis for the high efficiency and safety of solid waste disposal in cement kilns.

## 2. Materials and Methods

### 2.1. Raw Materials

The raw materials used in the present study were limestone, clay, and copper slag from a cement plant in Xianyang, Shaanxi. The chemical composition of each raw material is shown in Table 1. The heavy metal reagent used was chromium nitrate nonahydrate (Cr(NO_3_)_3_·9H_2_O) (analytical grade; Sinopharm Chemical Reagent Co., Ltd, Tianjin, China).

### 2.2. Sample Preparation

Blank samples were prepared according to the three major cement clinker ratio values IM = 1.309, SM = 2.580, and KH = 0.893. Based on this, the total amount of Cr was 0.4%, 0.6%, 0.8%, 1.0%, and 5.0% (with respect to pure heavy metal elements). For example, when the total amount of Cr was 0.4%, 3.1% of chromium nitrate nonahydrate (Cr(NO_3_)_3_·9H_2_O) was used. The ratio of raw materials is shown in Table 2. Appropriate amounts of the raw materials were mixed uniformly. The mixture along with an appropriate amount of distilled water was added in the pressure forming machine (Wanbo Instrument Co., Ltd, Hebi, China) to make a small test specimen of diameter 13 mm × 13 mm. The specimens were dried in an oven (Kewei Yongxing Instrument Co., Ltd, Beijing, China) at 105 °C. After drying, the test specimens were prefired in a muffle furnace (Capulet Limited, Britain) at 950 °C for 30 min, and then placed in a high-temperature furnace at 1450 °C for 30 min. The test specimens were then taken out and naturally cooled in air to obtain clinker samples.

### 2.3. Characterization and Analysis

Using the AVANCE 400 (SB) full digital NMR spectrometer of Bruker BioSpin Co., Ltd. in Switzerland the MAS-NMR analysis of ^29^Si (Bruker Biospin Co., Ltd, Switzerland) was used to analyze the clinker mineral phase structure. ^29^Si resonance frequency used was 79.49 Hz. Peak fitting software was employed to analyze the mineral phase content. The composition of the clinker mineral phase was investigated by XRD using the Japan Rigaku D/Max 2200 (Japan Rigaku Co., Ltd, Japan), where target was copper, and Kα rays with λ = 0.15418 nm were used. The morphology of clinker mineral phase was studied by SEM using the FEI Quanta 200 (FEI Limited, the U S), and EDS (FEI Limited, the U S) was used to analyze the distribution of Cr in clinker mineral phase.

## 3. Results

### 3.1. Effect of Cr on the Structure and Content of Mineral Phase

The solid nuclear magnetic resonance technology is useful for obtaining information about the structure of the material by analyzing the NMR spectra. Information regarding the number, position, width, and area of spectral lines can be inferred from the signal obtained by the NMR spectrum diagram. The number and position of spectral lines are reflected by the chemical shift; the changes in chemical shift indicate the changes in the chemical environment in which the nucleus is located, namely, the molecular structure of the material [19,20]. In the ^29^Si MAS NMR study, the chemical environment of ^29^Si is represented by Q^n^, where n (0–4) is the number of oxygen atoms shared by each silicon–oxygen tetrahedron and the adjacent silicon–oxygen tetrahedral. While Q^0^ represents an island-shaped [SiO_4_] tetrahedron with a chemical shift in the range of −68 to −79 ppm, Q^1^ represents the silicon–oxygen tetrahedron connected with one silicon–oxygen tetrahedron, that is, the silicon–oxygen tetrahedron is at the end of the straight chain in the dimer or hypermer, and the chemical shift is in the range of −76~−82 ppm. Further, Q^2^ represents the silicon–oxygen tetrahedron connected with two silicon–oxygen tetrahedrons, that is, the silicon–oxygen tetrahedron is in the middle of the straight chain or in the ring structure, and the chemical shift is in the range of −82~−88 ppm. Similarly, Q^3^ indicates the silicon–oxygen tetrahedron connected with three silicon–oxygen tetrahedrons, which has the branch of chain, double-chain polymerization structure, or layered structure with a chemical shift in the range of −88~−98 ppm. Finally, Q^4^ depicts four silicon–oxygen tetrahedrons forming a three-dimensional network structure with a chemical shift in the range of −98~−129 ppm [19,20,21,22].

Solid-state magnetic resonance technology is used to analyze the structure of solid materials with higher crystallinity or the structure of amorphous solid materials with lower crystallinity. Its unique quantitative analysis method has been applied to the research of cement-based materials. Cement clinker is mainly composed of dicalcium silicate (C_2_S) and tricalcium silicate (C_3_S). The ^29^Si MAS NMR analysis performed in the past [19,20,21,22] shows that the silicon–oxygen tetrahedrons in C_2_S and C_3_S are isolated structures, meaning that the chemical environment in which ^29^Si is located is Q^0^. The ^29^Si MAS NMR technique is, thus, suitable for analyzing the effect of Cr on the mineral phase structure of cement clinker.

Figure 1 is a ^29^Si MAS NMR spectrum of cement clinker fired with Cr. The chemical shift of the spectrum of the clinker with a Cr content of 0%–1% is −71.56 ppm, and that of the clinker with a Cr content of 5% is −71.17 ppm. In addition, as the Cr content varies from 0.4% to 0.8%, the NMR peak increases, and the peak becomes sharper. When the Cr content is 0.8%, the peak is at the highest magnitude, but it decreases at a content of 1.0%. The spectrum also becomes wider. Although the chemical shift of the ^29^Si MAS NMR spectrum has changed, the chemical environment of the ^29^Si is still Q^0^. The nature of the X group in the Si-O-X unit will affect the chemical shift of ^29^Si. While the Cr-O bond energy is weaker than that of Ca-O [22], the electronegativity of Cr^3+^ (1.66) is greater than that of Ca^2+^ (1.01). As a power-supplying group, Cr^3+^ replaces Ca^2+^, thus weakening the electron cloud density around ^29^Si, increasing the chemical shift, and shifting the spectral line to a low magnitude. For Cr content in the range of 0.4%–1.0%, the peak of the ^29^Si MAS NMR spectrum is relatively unchanged, smooth, and sharp. The widening phenomenon occurs when the Cr content reaches 5.0%, indicating that the high Cr content causes distortion of the crystal structure of the mineral phase [21], leading to generation of new mineral phases in the clinker.

In this ^29^Si MAS NMR analysis, the NMR spectrum peaks can be fitted. Different chemical shifts represent different structures in the sample. Deconvolution integration is performed to calculate the integrated area, and then the relative content of different structures can be estimated [23]. C_2_S has four crystal forms, namely, α-C_2_S, α’L-C_2_S, β-C_2_S, and γ-C_2_S. The chemical shifts corresponding to each crystal form are −70.7, −70.8, −71.4, and −73.5 ppm. C_3_S has 9 Si sites, the chemical shifts of which correspond to 7 positions of −69.2, −71.9, −72.9, −73.6, −73.8, −74.0, and −74.7 ppm, respectively [24]. Figure 2 depicts the peak fitting of the original spectrum using PeakFit software. This helped to determine the mineral phase composition represented by each peak position, and the area integral formula was used to calculate the relative content of each mineral phase. The relative percentages of C_3_S, β-C_2_S, a-C_2_S, and C_2_S, and the ratio of C_2_S to C_3_S were obtained, as shown in Table 3. With an increase in Cr content, the relative content of C_3_S gradually decreases, while that of C_2_S and β-C_2_S gradually increases. Further, the ratio of C_2_S to C_3_S shows an upward trend. The relative content of α-C_2_S reaches a maximum of 13.46% when the Cr content is 1.0%, and then decreases. When the Cr content is 5.0%, the relative content of C_3_S in the clinker reaches 32.56%, while that of C_2_S and β-C_2_S is 27.44% and 37.44%, respectively. The ratio of C_2_S to C_3_S is 0.6, probably due to the introduction of heavy metal Cr. Studies have shown that some heavy metal elements (Cr, Pb, Cd, Zn, etc.) act as stabilizers in the clinker calcination. They reduce the decomposition temperature of CaCO_3_ and the formation temperature of C_2_S, thus stabilizing β-C_2_S, and then consequently C_2_S, leading to an increase in C_2_S content [25]. In this study, since the addition of Cr has a positive effect on the cell size and the stability of the crystal form of β-C_2_S, the C_2_S content was seen to increase accordingly. The greater the amount of Cr, the more obvious was the effect on C_2_S.

### 3.2. Effect of Cr on Mineral Phase Formation

XRD analysis of Cr-doped cement clinker was performed. Figure 3 shows that the main mineral phases in the Cr-doped clinker are C_3_S, C_2_S, C_3_A, and C_4_AF. Compared with the blank sample, the C_2_S and C_3_S diffraction peaks for the five kinds of Cr-doped clinker are obvious. As the Cr amount increases, the C_2_S peak intensity and the number of peaks increase, indicating that a large amount of C_2_S is generated in the clinker. When the Cr content is 5.0%, the peak strength and the amount of C_3_S in the clinker decrease, indicating that C_3_S is generated to a lesser extent. Comprehensively, it is shown that Cr doping is beneficial to the stable formation of C_2_S and promotes the presence of C_2_S content, while an increase in the Cr amount inhibits the formation of C_3_S. In the five kinds of clinker, the C_3_A and C_4_AF peaks are not obvious, and the peak strength is weak, indicating that the C_3_A and C_4_AF contents are very small. In addition, when the Cr content is 5.0%, a weak intensity diffraction peak appears between the diffraction angles of 33° and 34°. The mineral phase of this diffraction peak is uvarovite, Ca_3_Cr_2_(SiO_4_)_3_, indicating that the addition of Cr affects the normal formation of crystal structure of clinker mineral phase, and then induces the formation of new crystal forms.

Figure 4 shows the analysis of the crystal structures of C_3_S, C_2_S, and uvarovite. C_3_S crystals are tightly packed with oxygen, calcium, and silicon entering the voids as octahedron and tetrahedron. The apex angles of [SiO_4_] tetrahedron are independent and exist in an isolated state. C_2_S is composed of [SiO_4_]^4^-tetrahedron and Ca^2+^. Compared with C_3_S, C_2_S has less O^2−^ in ionic composition, making the structure of C_2_S more stable. In the crystal structure of Ca_3_Cr_2_(SiO_4_)_3_, trivalent Cr ions, [CrO_6_], are formed with a coordination number of 6, connected with the isolated [SiO_4_] tetrahedron, forming some larger distorted cubic voids, and each apex angle is occupied by oxygen. Table 4 shows the radius, coordination number, and electronegativity of each ion. There are large differences in chemical structural parameters such as ionic radius, ligancy, and electronegativity between Cr^3+^, Ca^2+^, and Si^4+^ [26,27]. The ionic radius of Cr^3+^ lies between those of Ca^2+^ and Si^4+^, and its electronegativity is close to that of Si^4+^. From the point of view of the coordination number, Cr^3+^ replaces Ca^2+^, shifting the charge center of the ionic group, and causing the orientation of the silicon–oxygen tetrahedron and the coordination number of calcium to change. This means that the addition of Cr changes the mode of oxygen accumulation or accumulation density, and the cell parameters, resulting in the modification of silicate crystal shape and the generation of new calcium chrome garnet crystals.

### 3.3. Effect of Cr on Morphology of the Mineral Phase 

Figure 5a–c depict the SEM images of clinker with 0.8%, 1.0%, and 5.0% Cr, respectively. The SEM backscattering technique was used for these images. Point 1 in Figure 5a is a round grain, and may possibly be C_2_S. Point 2 is plate-like, and may be C_3_S. In all the images, the C_3_S and C_2_S mineral boundaries are clear and easy to distinguish. When the Cr content is increased to 5.0%, the C_2_S generation dominates, the aggregation state is strong, the mineral phase agglomerates greatly, and the C_3_S mineral phase is reduced. The high Cr content is beneficial to the stable formation of C_2_S and inhibits the formation of C_3_S. EDS spectrum analysis was performed on some microregions in the SEM images. The results are shown in Table 5. In Figure 5b, the Ca/Si atomic ratio in the EDS-1 microregion is 2.07, which is considered to be the C_2_S mineral phase, and the Ca/Si atomic ratio in the EDS-2 microregion is 2.67, which is considered to be the C_3_S mineral phase. The composition of EDS microregion and the content of Cr in mineral phase show that Cr is easily dissolved in C_2_S, and the content of Cr in C_2_S increases with the increase of Cr content. When the Cr content is 5.0%, its total mass in the microregion is about 3.98%.

## 4. Conclusions

The present study attempted to investigate the effect of Cr on the mineral phase formation in cement clinkers. The major conclusion was that Cr affects the formation of C_2_S and inhibits the generation of C_3_S. It does not, however, change the chemical environment of ^29^Si in the clinker mineral phase. It is still an isolated silicon–oxygen tetrahedron, with the Cr-O bond energy being weaker than that of Ca-O. The electronegativity of Cr^3+^ is also higher than that of Ca^2+^. Cr affects the orientation of the silicon tetrahedron and the coordination number of calcium, leading to defects in the crystal structure of the clinker mineral phase. Cr^3+^ replaces Ca^2+^ in the mineral phase lattice to form a new mineral phase, Ca_3_Cr_2_(SiO_4_)_3_. Cr plays a stabilizing role in the calcination of clinker to promote the formation of β-C_2_S. An increase in Cr content decreases the relative content of C_3_S and increases that of C_2_S. Cr easily dissolves in C_2_S, and C_3_S does not show the presence of Cr.

Thus, this study provides novel experimental and theoretical evidence for the solidification mechanism of heavy metals in the cement clinker mineral phase. In addition, in the actual production of sludge collaborative disposal by cement kiln, the content of Cr element in sludge will increase with an increase in sludge content, which will decrease the content of C_3_S in the clinker and increase the content of C_2_S. This degrades the quality and performance of the cement. Therefore, the pursuit for quantity should be avoided in sludge resource-based disposal, while the environmental risk from the Cr element is ignored, as well as the quality and performance risks of cement.

## Figures and Tables

**Figure 1 materials-13-01529-f001:**
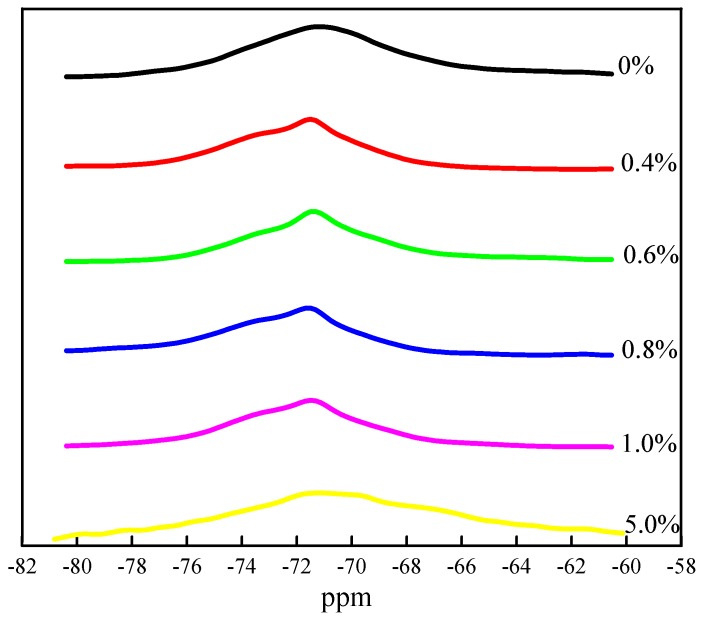
^29^Si MAS NMR spectrum of clinker with different Cr content.

**Figure 2 materials-13-01529-f002:**
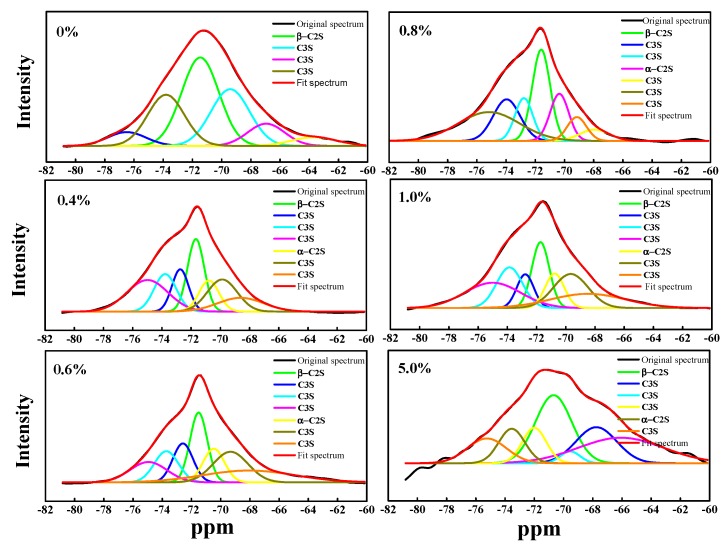
^29^Si MAS NMR spectrum after fitting peaks of different Cr doping clinker.

**Figure 3 materials-13-01529-f003:**
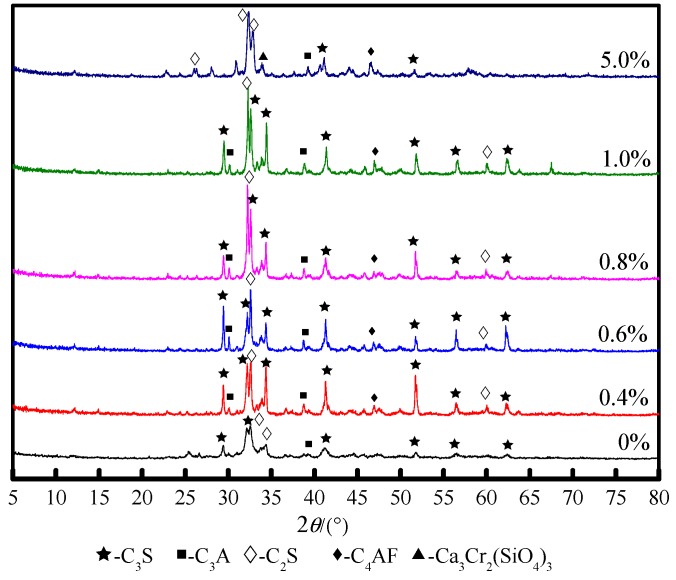
XRD patterns of clinker with varying Cr content.

**Figure 4 materials-13-01529-f004:**
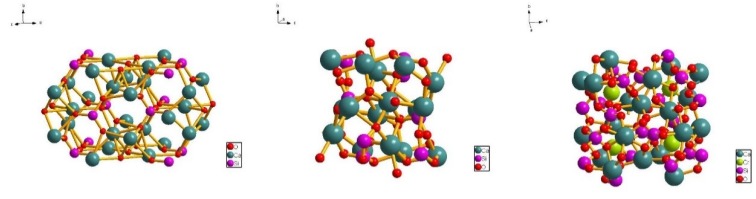
Crystal structure of silicate minerals.

**Figure 5 materials-13-01529-f005:**
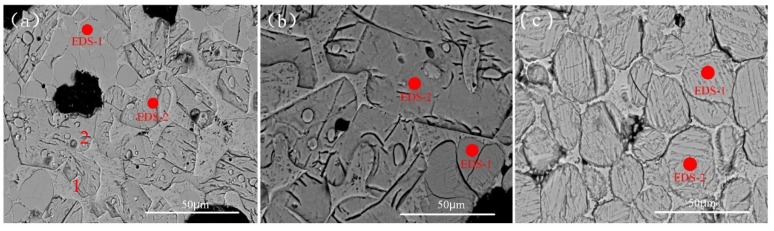
SEM spectrum of Cr-doped cement clinker. (**a**) with 0.8% Cr; (**b**) with 1.0% Cr; (**c**) with 5.0% Cr.

**Table 1 materials-13-01529-t001:** Chemical composition of the raw materials (wt %).

Composition	CaO	SiO_2_	Al_2_O_3_	Fe_2_O_3_
Limestone	56.80	0.28	0.12	0.05
Clay	14.52	48.25	10.28	4.60
Copper slag	6.55	30.17	8.93	48.04

**Table 2 materials-13-01529-t002:** Raw material ratio/wt %.

Raw Material	C0	C0.4	C0.6	C0.8	C1.0	C5.0
Cr	0	0.4	0.6	0.8	1.0	5.0
Limestone	68.73	68.73	68.73	68.73	68.73	68.73
Clay	28.92	28.92	28.92	28.92	28.92	28.92
Copper slag	2.35	2.35	2.35	2.35	2.35	2.35

**Table 3 materials-13-01529-t003:** Percentage of β-C_2_S, C_2_S, and C_3_S after peak fitting/wt %.

Content	0%	0.4%	0.6%	0.8%	1.0%	5.0%
β-C_2_S	36.2	19.43	19.03	23.51	23.12	27.44
C_2_S	36.2	29.36	30.21	36.13	36.58	37.44
C_3_S	63.8	70.64	70.72	63.87	63.42	62.56

Note: Relative percentage content refers to the percentage content of each component in the total silicon substance.

**Table 4 materials-13-01529-t004:** Ionic radius, coordination number, and electronegativity of ions.

Ion	Ionic Radius/pm	Electronegativity	Ligancy
Ca^2+^	100	1.0	6
Si^4+^	40	1.9	4
Cr^3+^	69	1.6	6

**Table 5 materials-13-01529-t005:** EDS analysis of Cr-doped clinker microregion.

Number		Mass Fraction/%				
		O	Al	Si	Ca	Cr
a	EDS-1	50.65	1.51	14.48	32.02	1.35
	EDS-2	53.90	1.30	11.63	32.07	——
b	EDS-1	53.51	1.29	14.28	29.57	1.34
	EDS-2	53.22	1.02	12.11	32.19	——
c	EDS-1	54.33	1.51	14.15	29.03	3.98
	EDS-2	53.98	1.77	14.02	29.13	4.10

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
