# Peer review of "Effect of Cr on the Mineral Structure and Composition of Cement Clinker and Its Solidification Behavior"

_materials, 2020, doi:10.3390/ma13071529_

Round 1
Reviewer 1 Report
I believe that the article does not contribute anything new to the state of current knowledge. Neither research methods nor the results are innovative. The authors themselves stated that "The major conclusion was that Cr affects the formation of C2S and inhibits the generation of C3S. It does not, however, change the chemical environment of 29 Si in the clinker mineral phase. " Yet, it is the editor's decision whether such an article should be published in the journal "Materials" If the decision is yes, then I have a few minor comments:
- Line 81-82 - It seems to me that the name of the company as surreptitious advertising should not be placed there, especially since it adds nothing to the information from the research
- The description presented in chapter 3.1 lines 93-103 is not clear. There is a lot of information that is difficult to understand. I suggest a different form of presenting this information
- Figures are generally poorly described. Descriptions of axes are missing.
Reviewer 2 Report
The aproach is intereting and well presented.
Couldn't find more recent references?
Reviewer 3 Report
The paper describes the effect of Cr on mineral structure and composition of cement clinker. The research is new and could be useful for research community. The following comments are suggested:
- The Cr concentration in the sample is set to vary from 0 to 5%. How did the authors arrive at this range?
- From the XRD analysis, it is shown that higher concentration of Cr can result in lower Cr3S and a dominating Cr2S phase. In the context of sludge disposal, what does this signify? This needs to be explained in the manuscript.
- The significance of the work needs to be highlighted in the abstract section.
- In Fig. 2, the graph legends should be in English.
Reviewer 4 Report
The manuscript reports on the XRD and Magic angle Si-NMR study of various compositions of clinker with increasing addition of chromium. As indicated in the introduction the valorisation of industrial sludge rich in heavy metal such as chromium is important task and the topic is interesting. The experimental work is well conducted but the writing makes difficult the reading. The paper starts with Si MA-NMR study, a local analyse and the XRD data are given after. Some background is also missing. The paper should be published after the following points have been improved.
First the authors must summarize the information on the different phases that can be formed and give correspondence between Cement’ experts community jargon (C2S,C3S, ...) and crystallographic names (alite, belite,...), according XRPD labelling. Qn content for each structure must be also explained.
The amount of chromium is expressed sometime in Cr2O3 oxide, sometime in Cr element or in Cr3+. A common description must be used or if two are needed, both must be indicated.
To make more visible the progressive local structure changes the spectra of Fig. 1 must be shifted vertically. For the same reason the spectra of Fig. 2 must be shown in vertical order:
0 0.8
0.4 1
0.6 5
Its a pitty that the authors do not perform Raman study. Under green laser excitation Cr-O bond led to the Raman (pre)Resonance Spectroscopy; thus the technique is very efficient (see. e.g. J. Raman Spectroscopy 2003, 34, 290)
